# Tau-Targeted Therapeutic Strategies: Mechanistic Targets, Clinical Pipelines, and Analysis of Failures

**DOI:** 10.3390/cells14191506

**Published:** 2025-09-26

**Authors:** Xinai Shen, Huan Li, Beiyu Zhang, Yunan Li, Zheying Zhu

**Affiliations:** Division of Molecular Therapeutics and Formulation, School of Pharmacy, The University of Nottingham, Nottingham NG7 2RD, UK; xinai.shen@nottingham.ac.uk (X.S.); paxbz1@exmail.nottingham.ac.uk (B.Z.); paxyl9@exmail.nottingham.ac.uk (Y.L.)

**Keywords:** tau protein, Alzheimer’s disease, tauopathies, clinical trials

## Abstract

Tau protein, a neuron-enriched microtubule-associated protein encoded by the *MAPT* gene, plays pivotal roles in microtubule stabilisation, axonal transport, and synaptic plasticity. Aberrant post-translational modifications (PTMs), hyperphosphorylation, acetylation, ubiquitination, oxidative stress and neuroinflammation disrupt tau’s normal functions, drive its mislocalization, and promote aggregation into neurofibrillary tangles, a hallmark of Alzheimer’s disease (AD) and related tauopathies. Over the past two decades, tau-targeted therapies have advanced into clinical development, yet most have failed to demonstrate efficacy in human trials. This review synthesises mechanistic insights into tau biology and pathology, highlighting phosphorylation and acetylation pathways, aggregation-prone motifs, and immune-mediated propagation. We analyse the current therapeutic landscape, including kinase and phosphatase modulators, O-GlcNAcase inhibitors, aggregation blockers, immunotherapies, and microtubule-stabilising agents, while examining representative clinical programs and the reasons underlying their limited success. By combining mechanistic understanding with clinical experience, this review outlines emerging opportunities for rational treatment development, aiming to inform future tau-targeted strategies for AD and other tauopathies.

## 1. Introduction

Alzheimer’s disease (AD) is the leading cause of dementia, affecting more than 40 million people worldwide [1]. Current therapies, including acetylcholinesterase inhibitors, the NMDA receptor antagonist memantine and two monoclonal antibodies (mAbs), provide only modest symptomatic relief, delaying cognitive decline by a few months [2,3,4,5]. Despite decades of intensive research, numerous disease-modifying approaches, particularly immunotherapies targeting amyloid-β (Aβ) or tau, have shown target engagement but failed to deliver consistent clinical benefits [2,6]. Because of these limitations, novel treatment strategies are urgently needed. Tau is a microtubule (MT)-associated protein essential for cytoskeletal stability and axonal transport, has gained increasing attention as a target [7]. Abnormal post-translational modifications and aggregation of tau underlie neurofibrillary tangles, which correlate more closely with cognitive decline than amyloid burden [8,9,10]. Moreover, mutations and splicing alterations of the *MAPT* gene highlight tau dysfunction as a direct cause of neurodegeneration [11,12]. In recent years, therapies targeting tau protein, including immunotherapy, small-molecule therapies, antioxidant enzymes, and gene therapy, have entered clinical development, expanding the therapeutic prospects for AD and other tauopathies [13,14]. This review will summarise potential targets for the tau protein and strategies currently under clinical investigation, assessing their translational potential and the challenges that remain in achieving effective disease modification. Information was primarily sourced from ClinicalTrials.gov, including keyword searches “tau”, “*MAPT*”, “Alzheimer’s disease”, “Progressive supranuclear palsy”, “frontotemporal dementia”, “Parkinson’s disease” and “Lewy body dementia”. Additional details were extracted from company websites, Alzforum, PubMed, and conference proceedings when relevant. The search covered the period from July 2015 to July 2025 and included compounds that initiated Phase I–III clinical trials within this timeframe, as well as programs discontinued after clinical trials. Results were sorted by molecule type and clinical status. We recognise that many discovery-stage and preclinical candidates remain undisclosed, and these have not been captured in this overview. All tables and narrative presentations reflect the information available at the time of publication.

## 2. Physiological and Pathological Mechanisms of Tau

Tau is a neuron-enriched MT-associated protein encoded by *MAPT*, with detectable expression in oligodendrocytes and low-level expression in other glia [15]. In the adult human central nervous system (CNS), alternative splicing of exons 2, 3 and 10 produces six brain-specific tau isoforms with baseline differences in MT affinity and interaction networks due to their N-terminal inserts and 3R/4R repeat composition [16,17]. Beyond these six CNS isoforms, *MAPT* generates additional tissue- and development-restricted isoforms, notably the high-molecular-weight “Big Tau” defined by inclusion of exon 4a (~110 kDa), enriched in the peripheral nervous system (PNS) and certain CNS tracts [18]. In physiological conditions, tau binds the MT lattice to stabilise tracks, regulate interfilament spacing, gate kinesin access for axonal transport, couple transport to translational control at synapses, and help maintain nuclear chromatin architecture [19,20,21]. Studies indicate that disease begins with upstream shifts in post-translational modification (PTM) homeostasis, synaptic stress and kinase imbalance, compounded by reduced O-GlcNAcylation, which drives persistent hyperphosphorylation [22,23,24]. Overmodified tau accumulates in somatodendritic regions due to axonal sorting and reduced MT affinity at the axon initial segment (AIS) diffusion barrier [25]. Consistent with AIS-dependent sorting, tau isoforms lacking the N-terminal insertion are efficiently targeted to axons, while the longest 2N4R isoform exhibits partial somatodendritic retention [26]. Specific acetylation impairs ubiquitin proteasome-mediated clearance, thus enhancing oligomerisation and accumulation of filamentous tau [27,28]. Seed components cross synapses through vesicle pathways and then amplify through innate immune signalling cascades, thereby enhancing PTM dysregulation [29,30]. Failures in axonal transport, mitochondrial homeostasis, and synaptic transmission produced in this manner produce regional neurodegenerative responses [31]. Heterogeneity at the epitope and isoform levels is a defining characteristic of tau biology, directly impacting translation. Three dimensions are particularly relevant to therapeutic design and trial interpretation: isoform diversity (i.e., 3R/4R composition and region-specific expression), conformer/strain variation (i.e., assembly-dependent epitopes with distinct seeding behaviours), and compartmentalisation (i.e., axonal-dendritic mislocalization and extracellular proliferation versus intracellular reservoirs) [32,33,34]. These dimensions dictate which epitopes are exposed, which species can be neutralised in vivo, and which therapeutic approaches, such as antibodies, oncogenes, or vaccines, are most likely to bind to disease-associated tau [35,36].

### 2.1. Physiological Roles of Tau in Neurons

Tau protein is encoded by the *MAPT* gene on chromosome 17q21.31 [12]. As a member of the MT-associated protein family, tau is primarily expressed in neurons, where it stabilises and regulates axonal MTs [37]. It is transcribed as a single pre-mRNA and processed by alternative splicing to yield six major isoforms by the presence or absence of N-terminal inserts and the number of MT-binding repeats in the C-terminus [38]. Specifically, exons 2 and 3 encode N-terminal inserts generating 0N, 1N, and 2N variants, while exon 10 inclusion produces 4 MT-binding-repeat (4R) isoforms and its exclusion generates three-repeat (3R) isoforms [39,40]. Tau’s primary physiological function is stabilising and promoting the assembly of axonal MTs [41], thereby supporting neuronal morphology, axonal transport and cytoskeletal integrity [42]. Isoform context frames epitope availability even under physiological conditions. 4R isoforms show higher MT affinity and dwell time on the lattice than 3R species, while N-terminal inserts modulate interfilament spacing and protein–protein interactions [43,44]. These intrinsic differences influence which repeat-domain epitopes are solvent-exposed and help explain region-specific vulnerability. This physiological baseline becomes clinically actionable once isoform ratios shift in disease, because epitope choice for antibodies and vaccines and splice-modulating strategies for antisense oligonucleotides (ASOs) can be aligned to the predominant isoform pool [39,45,46,47]. Besides cytoskeletal support, tau regulates cargo transport, recruits signalling molecules, and modulates translation by binding ribosomes and repressing protein synthesis, particularly in synaptic contexts where plasticity depends on local translation [48,49,50]. Studies implicated tau in the regulation of long-term memory, habituation, sleep–wake cycles, and synaptic plasticity via effects on long-term potentiation/depression (LTP/LTD) [51,52,53,54].

In addition to its MT-stabilising role, tau exhibits diverse cellular functions that extend its regulatory influence. Tau regulates motor protein dynamics by forming concentrated microdomains along axonal MTs, thereby differentially affecting kinesin and dynein mobility, enabling precise spatial control and cargo delivery [55]. Besides the cytoskeleton, tau also participates in synaptic plasticity and neuronal communication by regulating dendritic spine density and synaptic strength mechanisms. For instance, tau deficiency impairs LTP and dendritic spine density, highlighting its essential role in cognitive performance [56,57]. Furthermore, tau supports neurogenesis and neuronal maturation, as evidenced by its involvement in dendritic development and granule cell migration [58]. Tau also performs protective nuclear functions, safeguarding genomic integrity under stress conditions and preventing chromosomal instability by maintaining pericentromeric heterochromatin organisation [59,60]. In addition, tau is involved in regulating iron homeostasis and mitochondrial transport, which contributes to cellular metabolism and oxidative balance [61,62,63].

Genetic studies using tau-deficient mice have revealed the complexity of tau’s physiological roles. Although tau knockout mice develop without severe mental defects due to other tubulin compensations, tau deficiency results in hyperactivity, reduced LTP, altered fear-conditioning behaviour, disrupted sleep patterns with increased wakefulness and reduced non-rapid eye movement sleep, anxiety-related behaviour, muscle weakness, impaired peripheral glucose metabolism, and pancreatic function [64,65,66,67]. These phenotypes demonstrate that tau participates in neural and behavioural regulation physiologically.

### 2.2. Tau Dysregulation and Pathological Transitions

In disease, isoform and conformer heterogeneity are amplified. AD features mixed 3R/4R filaments, whereas progressive supranuclear palsy (PSP)/corticobasal degeneration (CBD) are 4R-dominant and Pick’s disease is 3R-predominant [68]. Conformer differences map onto seeding kinetics and regional spread, such that antibodies raised to N-terminal linear epitopes may fail to bind aggregation-competent cores, whereas microtubule-binding region (MTBR)-directed epitopes remain accessible across seeds [69]. Therapeutically, isoform composition and fold-specific exposure act as gatekeepers for epitope targeting.

#### 2.2.1. Phosphorylation: A Central Driver of Tauopathy

Pathologically phosphorylated tau severely impairs synaptic integrity through multiple mechanisms, including disruption of dendritic spine morphology, synaptic vesicle release, receptor trafficking, and mitochondrial function, resulting in cognitive deterioration [57,70]. Under pathological conditions such as AD and other tauopathies, phosphorylation homeostasis is shifted away from dephosphorylation. The principal tau phosphatase protein phosphatase type 2A (PP2A) exhibits reduced catalytic activity and subunit expression, while multiple kinase axes are engaged [71,72,73,74], resulting in tau becoming abnormally hyperphosphorylated. Tau phosphorylation in mediated by several kinase classes. Proline-directed protein kinases (PDPKs), including glycogen synthase kinase-3β (GSK-3β), which phosphorylates tau on serine or threonine residues that are followed by a proline residue. Within the non-PDPK family are tau-tubulin kinases 1 and 2, casein kinases 1 and 2, dual-specificity tyrosine-phosphorylated and -regulated kinase 1A (DYRK1A), phosphorylase kinase, Rho kinase, protein kinase A (PKA), protein kinase B (PKB)/Akt, protein kinase C (PKC) and protein kinase novel (PKN) [75].

To date, more than 85 phosphorylation sites have been identified on tau, primarily on serine and threonine residues [38,76]. Before the overt tangle formation, some early detectable signal rises, including pTau181, pTau202/pTau205, pTau217, and pTau231, serving as a biochemical shift in the kinase-phosphatase equilibrium that favours phosphorylation [77,78,79], which eventually leads to approximately a three-fold increase in phosphate groups than in the normal adult brain [80,81]. These sites have been widely used as a diagnostic biomarker in both human brain tissues and cerebrospinal fluid (CSF) [82,83,84]. Tau phosphorylation at Ser262 in the KXGS motifs of MT-binding repeats reduces MT binding, promoting detachment from axonal MTs and somatodendritic/synaptic mislocalization [85,86]. Mislocalized hyperphosphorylated tau exposes the PHF6 (VQIVYK) and PHF6* (VQIINK) hexapeptides that nucleate β-sheet assembly, driving the nucleation–elongation of paired helical filaments (PHFs) and straight filaments [87,88]. Subsequently, PHFs form into neurofibrillary tangles (NFTs), which are hallmarks of tauopathies [14].

In this maturation window, pTau396 and pTau422 index later-stage tangle pathology, whereas pTau199, which is used in several legacy CSF assays, tracks with the broader phospho-load but is outperformed by pTau217 for the early stage of AD [89,90,91]. In addition to AD, aberrant tau phosphorylation is implicated in other tauopathies such as PSP, CBD, and frontotemporal dementia with Parkinsonism linked to chromosome 17 (FTDP-17) [92]. Across tauopathies, the process of phosphorylation development is common, but isoform usage and epitope kinetics are different: AD contains mixed 3R or 4R PHFs and NFTs with early pTau181, pTau217 dynamics, while PSP or CBD are 4R-predominant with distinct phosphorylation signatures [93,94]. Because early phospho-epitopes (e.g., pTau217/231) index upstream kinase imbalance but not fibril burden, production-lowering agents (ASOs/siRNA) are best monitored by pTau panels paired with regional tau-PET slopes. In contrast, antibodies or vaccines directed at MTBR or disease-enriched conformational epitopes require readouts that track aggregate load, such as CSF MTBR-tau243 and tau-PET [95].

#### 2.2.2. Acetylation and Crosstalk with Protein Clearance Pathways

Tau acetylation is a post-translational modification that may modulate normal tau function and lead to abnormal tau aggregation in AD and related tauopathies. Mechanistically, lysine acetylation is catalysed by p300/CBP using acetyl-CoA, while histone deacetylases (HDAC) and sirtuins (SIRT) deacetylate distinct sites of tau [96]. In vitro, tau can also autoacetylate in the presence of acetyl-CoA, further contributing to total acetyl load. These enzyme-level inputs initiate the acetylation cascade [97].

At the chemical/biophysical layer, acetylation neutralises the positive charge on targeted lysine within the MT-binding repeats, weakening electrostatic interactions with the MT lattice and impairing tau–MT binding [98]. Increasing acetylation at Lys163 [99], Lys174 [100], Lys280 [27], Lys281 [101,102], and Lys369 [103], the interaction between tau and MTs was impaired [104]. Lys174 and Lys280 have been specifically associated with insolubility and the accumulation of pathogenic tau forms. It has been demonstrated that elevated levels of acetylated tau were present in patients at early and moderate Braak stages of tauopathy, suggesting that acetylation may play a role in the early stages of AD pathology [105]. At the proteostasis level, acetylation competes with ubiquitination on the same lysines, therefore preventing proteasomal degradation and stabilising pathologic tau. Ubiquitin–proteasome system (UPS) flux declines as ubiquitin acceptor sites are masked, and chaperone-mediated autophagy recognition is hindered, slowing lysosomal clearance [106]. For example, Lys174 acetylation with an increase in p300/CBP and a decrease in SIRT, blocks degradation and exacerbates behavioural and synaptic deficits in tauopathy models [96]. Although acetylation within the repeat domain facilitates oligomerisation and fibrillization, it is not uniformly detrimental. Within KXGS motifs, Lys321 and Lys353 may represent a natural protective mechanism against tau aggregation and therefore a potential therapeutic target [107]. These different acetylation sites point to therapeutic selectivity as essential. Acetylation within the repeat domain not only weakens MT-binding but stabilises seed-competent folds that mask some linear epitopes while exposing others within MTBR (e.g., HVPGG-proximal surfaces) [108]. This helps explain why N-terminal antibodies have shown weak biomarker–clinical linkage in several trials, whereas MTBR-focused designs demonstrate more consistent biochemical engagement. Vaccine epitopes that mimic disease-enriched assemblies may similarly improve specificity over total-tau immunogens.

#### 2.2.3. Oxidative Stress-Induced Modifications of Tau

Oxidative stress is another common feature of tauopathies and has multiple sources. These include mitochondrial reactive oxygen species (ROS) generated by Ca^2+^-overloaded neurons, NADPH oxidase (NOX2)-driven superoxide in activated microglia/neurons, and metal-catalysed Fenton chemistry producing hydroxyl radicals [109,110,111]. These oxidants can shift the kinase-phosphatases. Calpain activation by Ca^2+^ overload cleaves p35 to p25 and activates CDK5. Oxidative stress activates GSK-3β and extracellular signal-regulated kinase (ERK) while inhibiting AKT, thus favouring tau phosphorylation at Thr231 and Ser396. In addition, the principal tau phosphatase PP2A is inhibited by oxidative demethylation or oxidation and regulatory dysfunction [112,113]. ROS directly modify tau, biasing structure and clearance, Cys291 and Cys322 promote intramolecular disulfides that nucleate oligomers, nitration of tyrosines reduces MT binding and fosters aggregation, and lipid peroxidation adducts such as 4-hydroxy-2-nonenal (4-HNE) form Michael adducts on lysines, further weakening MT affinity and inhibiting the proteasome [114,115,116].

#### 2.2.4. Neuroinflammation as a Catalyst of Tau Propagation

Neuroinflammation in tauopathies is caused by convergent pathological triggers that produce a self-perpetuating cycle of immune activation and tau dysfunction. Misfolded and oligomeric tau initially acts as an intracellular damage-associated molecular pattern, while synaptic dysfunction generates excess glutamatergic drive and proteinopathy-induced mitochondrial distress, which elevates inflammatory markers [117]. In response, activated microglia undergo metabolic reprogramming and shift the kinase–phosphatase regulatory equilibrium by activating p38 MAPK, ERK1/2, and GSK-3β pathways while functionally suppressing PP2A, thus promoting hyperphosphorylation at key tau epitopes [118,119,120]. Furthermore, activated microglia phagocytose pathological tau and subsequently package and secrete tau seeds within exosomes, creating vehicles for trans-cellular spread that bypasses normal clearance mechanisms, which facilitates tau pathology propagation [121]. Genetic or pharmacological blockade of exosome biogenesis using neutral sphingomyelinase-2 inhibitors significantly limits trans-synaptic propagation in mouse models, establishing neuroinflammation as a critical driver of tau propagation [122]. Therefore, the inflammatory mediators promote tau post-translational modifications and cellular mislocalization, leading to enhanced seeding competency and propagation efficiency [123]. Microglial vesicle trafficking and LRP1-mediated neuronal uptake shift a substantial fraction of seed-competent tau to the extracellular/interstitial space, where antibodies and vaccines can neutralize and opsonize assemblies. Yet the bulk of tau mass remains intracellular, arguing for ASOs, intrabodies, or nanobodies to reduce the neuronal reservoir [124]. This intra- vs. extracellular partitioning underlies the mechanistic complementarity of combination regimens and clarifies why some extracellular antibodies show limited clinical translation despite plausible biology.

## 3. Therapeutic Strategies Targeting Tau

### 3.1. Modulating Enzymes Governing Tau Post-Translational Modifications

In AD, tau accrues several abnormal PTMs, including hyperphosphorylation, acetylation, and ubiquitination, which weaken MT-binding, mislocalize tau to dendrites and synapses, and prime NFT assembly. Given the central role of hyperphosphorylation in tau pathology, considerable effort has been directed toward targeting tau phosphorylation as a therapeutic strategy. Several approaches aim to inhibit the kinases responsible for pathological phosphorylation or enhance the activity of tau phosphatases. Suppressing pathological phosphorylation by inhibiting kinases or activating phosphatases is a major strategy. Among kinase inhibitors, GSK-3β and CDK5 remain the most explored nodes. The GSK-3β inhibitor tideglusib reduced phosphorylated tau in a mouse model but failed to deliver clinical benefit in randomised phase II AD trials, highlighting target engagement and time-window limitations [125,126]. In contrast, inhibition of Fyn/Src proteins with saracatinib (AZD0530) achieved ample brain penetration and biomarker-target engagement but did not translate into clinical efficacy, suggesting that upstream synaptic signaling modulation alone may be insufficient [127]. Similarly, p38α MAPK inhibitors, such as Neflamapimod, demonstrated encouraging biomarker changes in exploratory studies but failed to demonstrate significant cognitive benefits [128]. These findings, taken together, suggest that while distinct kinase nodes regulate tau phosphorylation, their effects are limited in distinct ways: tideglusib is limited by disease stage sensitivity, saracatinib by downstream redundancy, and p38α inhibition by incomplete mechanistic selectivity. On the phosphatase side, PP2A can be pharmacologically active. Sodium selenate increases PP2A activity and dephosphorylates tau. It also improved cognition in preclinical AD models [129].

Another strategy associated with reducing phosphorylation is to regulate tau O-glycosylation. Because O-GlcNAcylation on Ser and Thr sites antagonises phosphorylation, O-GlcNAcase (OGA) inhibitors elevate tau O-GlcNAc to dampen multisite hyperphosphorylation and fibrillization. Multiple brain-penetrant OGAs entered the clinic. Ceperognastat illustrates central target engagement and robust OGA occupancy, but failed its phase II primary endpoint in early AD [130]. ASN90 advanced toward phase II [131]; BIIB113 completed phase I but was discontinued in 2025 [132]. The emergence of LY-3372689 demonstrates Lilly’s aggressive attempts to study tau-targeted drugs, especially OGA inhibitors. This mechanism remains compelling, warranting further clinical evaluation. Overall, the mechanism remains compelling, but duration, dose and stage selection likely determine clinical trials’ success.

Another strategy is to correct the acetylation axis. Small-molecule p300 inhibitors such as salsalate reduced ac-Lys174 and total tau in vivo [133]. Conversely, site-specific acetylation within KXGS motifs is perhaps protective, underscoring site-dependence and the need for selective HDAC pathway [134]. Beyond enzymes, Selenoprotein W was shown to promote tau ubiquitination and inhibit Lys281 acetylation, thereby restoring UPS-dependent clearance [135]. Overall, the balance among kinase, phosphatase, and O-GlcNAc governs the initiation of tau expression level, while acetylation–ubiquitination crosstalk regulates its stability and clearance level. These key steps can all be used as drug design targets.

### 3.2. Inhibitors of Tau Aggregation

Aggregation is initiated when post-translational and environmental cues expose the β-sheet–forming hexapeptides PHF6 (VQIVYK) and PHF6* (VQIINK) in the repeat domain, creating nuclei that template cross-β assembly. Tau is further concentrated into condensates that mature into oligomers, which form protofibrils that become paired helical filaments [88]. Based on different mechanisms, inhibitors are divided into four types: phenothiazine derivatives, natural polyphenols, peptide-based inhibitors, and metal chelators. Among small-molecule inhibitors, phenothiazine derivatives have been extensively studied for their ability to prevent tau aggregation. Phenothiazines bind aggregation-prone interfaces within the repeat domain to disrupt inter-strand hydrogen bonding, thereby destabilising fibrils and slowing elongation. These compounds, particularly methylene blue and TRx0237, both intercalate at cross-β interfaces, reducing the thioflavin signal [136,137]. Despite initial preclinical success, methylene blue failed to meet primary cognitive endpoints, with post hoc monotherapy signals remaining controversial [138]. These issues may illustrate the brain exposure and endpoint sensitivity. Phenothiazines, therefore, map to the fibril-remodelling blockade tier, and future development based on optimised CNS pharmacokinetics and biomarker-mediated designs.

Natural polyphenols have been widely investigated for their anti-aggregation properties [139]. For example, epigallocatechin gallate (EGCG) and curcumin both have this effect. EGCG is a green tea catechin, engages multiple hot spots on tau fibrils to remodel mature cross-β architecture into off-pathway, non-toxic oligomers, while impeding further elongation. This intervention reduces tau propagation and lowers toxicity readouts, but shifts the ensemble rather than preventing nucleation [140]. Current research focuses on formulation strategies to enhance their delivery efficiency into the brain.

Peptide inhibitors target key aggregation-prone motifs in tau, competitively bind PHF6 and PHF6* raise the nucleation barrier and cap growing filament ends, thereby suppressing primary nucleation and early elongation [141]. Because native L-peptides are protease-sensitive and blood–brain barrier (BBB) limited, the second-generation designs use lipidation and intranasal delivery to preserve target occupancy in the CNS [142]. These peptides act upstream of fibril remodellers and are best positioned for early-stage disease.

Metal ions such as zinc and copper can accelerate tau aggregation through direct coordination to histidines and cysteines and by promoting oxidative crosslinking [143]. Therefore, chelators lower the effective cofactor activity, attenuating nucleation and off-pathway oxidative stabilisation of oligomers. Clioquinol and PBT2 reduced tau and Aβ aggregation and toxicity in models but failed to reveal consistent clinical benefit in AD, largely due to specificity and safety constraints [144,145]. Consequently, metal modulation now serves as an adjunct strategy, ideally coupled with direct aggregation blockers and redox control.

### 3.3. Enhancing Tau Clearance via Immunotherapy

Pathogenic tau is thought to spread as extracellular assemblies that enter neurons via heparan-sulfate-dependent macropinocytosis and related pathways. Thus, immunotherapies can lower PHF and NFT accrual and propagation by blocking uptake and routing immune complexes to microglial endolysosomes via Fcγ receptors. These approaches intersect with spreading nodes by neutralizing seed-competent extracellular tau and routing immune complexes to Fcγ receptor-dependent lysosomal clearance in microglia, thereby reducing exosome-supported spreading and HSPG/LRP1-mediated neuronal uptake [146].

Active immunotherapy strategies aim to elicit an endogenous immune response against pathological tau species through vaccination. Although fewer in number compared to passive approaches, tau vaccines offer potential advantages in terms of long-term efficacy and lower treatment burden. JNJ-2056 (ACI-35030) is a liposomal phosphorylated tau vaccine that is in randomised mid-stage testing, reflecting its positioning to intercept propagation before overt tangle spread. By design, antibody output is biased toward late phospho-epitopes that track with fibril maturation, aiming for long-lived neutralisation without frequent infusions [147]. In contrast, GV1001 (tertomotide) is a telomerase peptide vaccine repurposed to AD and PSP, which has anti-inflammatory and antioxidant actions and has entered phase II studies, but its putative benefit relates to neuroprotective immunomodulation rather than direct anti-tau binding [148,149].

Passive approaches have the ability to target different stages of tau through pairing neutralisation with Fc-driven microglial clearance [150]. E2814 (etalanetug) is a humanised IgG1 against the MTBR, targeting the MTBR HVPGG motif and is bi-epitopic on 4R and mono-epitopic on 3R tau, aligning with propagating seeds [151]. In AD and primary tauopathies producing up to 75% and 50% reductions in CSF MTBR-tau243 and pTau217 over 2 years in dominantly inherited AD, with tau-PET trends to decline versus natural history [152]. It is being advanced in phase II/III trials. Some other mAbs illustrate orthogonal epitope strategies now in early clinical phases. LuAF-87908 is a humanised IgG1 antibody directed against pTau, while MK-2214 specifically targets the Ser413 phosphorylation site, a residue implicated in tau pathology [153,154]. APNmAb005 is designed to recognise high molecular weight and aggregated tau species that are thought to be particularly neurotoxic. VY-7523 binds to the C-terminal domain of tau, potentially interfering with tau-mediated intracellular interactions or aggregation [153]. Therapeutic programs target tau at different stages: antibodies against early phospho-epitopes aim to intercept initiation, while MTBR- and oligomer-selective monoclonal antibodies are designed to limit ongoing spread and to promote microglial lysosomal degradation of pathogenic tau [155]. Although more candidates have emerged for immunotherapy, clinical trials have shown that targeted intervention cannot be fully translated into clinical benefits. Poor antibody exposure in the brain parenchyma is one of the main limitations. The interstitial concentration of conventional intravenous IgG is low, making it difficult to maintain targeted occupancy at the synaptic level [156]. The clinical results of mAb semorinemab did not reach the primary efficacy endpoint of reducing the decline in the clinical dementia score (CDR-SB), nor did they reach the two secondary focus points of evaluating ADAS-Cog13 and ADCS-ADL performance [157,158]. In order to increase brain exposure, receptor-mediated transcellular transport (RMT) is a current research direction. This non-invasive strategy binds to specific receptors on the surface of endothelial cells and can transport large molecules across the BBB. Furthermore, most current anti-tau antibodies target N-terminal epitopes, such as Biogen’s gosuranemab and AbbVie’s ABBV-8E12 (which recognises an epitope near the N-terminus) [13]. These antibodies do not specifically target pathogenic tau, such as phosphorylated tau, failing to maximize therapeutic efficacy and often resulting in clinical failure [13,159]. Therefore, the development of antibodies needs to be more targeted at pathogenic tau protein. Although tau antibodies are typically formulated with IgG4 or Fc-reduced antibodies to reduce microglial hyperactivation, infusion reactions and ADA still need to be monitored to avoid suspension due to side effects like AN-1792 [160,161].

### 3.4. Microtubule-Stabilising Agents as Compensatory Strategies

In tauopathies, hyperdynamic MTs and transport failure arise as the tau function is compromised, providing a mechanistic rationale for MT-stabilising agents (MSAs) that restore dynamics toward a healthy regime. [162]. Among the various MSAs investigated, paclitaxel demonstrated initial promise but proved unsuitable for neurotherapeutic applications due to its limitation to BBB penetration and significant peripheral toxicity [163,164]. In contrast, epothilone D has emerged as the most clinically relevant candidate, distinguished by its superior BBB permeability and favourable safety profile at therapeutically relevant doses. Preclinical validation in tau transgenic mouse models has demonstrated that epothilone D treatment produces multiple beneficial effects, including enhanced MT density, improved axonal transport dynamics, reduced tau pathology burden, and significant amelioration of cognitive deficits [165,166,167].

Although the taxane analog TPI-287 has entered Phase I/II clinical trials for AD and PSP/CBS, hypersensitivity reactions have occurred, and no significant clinical benefit has been demonstrated [168]. Among these agents, paclitaxel has poor brain penetration, epothilone D has demonstrated CNS activity but has uncertain long-term tolerability, and TPI-287 achieves brain exposure but raises concerns about peripheral toxicity [169]. These differences highlight that BBB permeability, isomer selectivity, and long-term tolerability are key differentiating factors among MSAs.

Despite these promising preclinical results, several challenges remain in translating MSAs to clinical applications. The primary concern involves the inherent non-selectivity of current agents, which bind to all tubulin-containing MTs. This broad activity profile raises concerns about potential off-target effects in peripheral tissues, particularly given that many MSAs were originally developed as anti-cancer agents with established toxicity profiles. Additionally, the long-term consequences of chronic MT stabilisation in the nervous system remain incompletely characterised. Current efforts focus on developing neuron-selective MSAs with improved safety profiles [170]. Moreover, the complementary mechanism of action suggests potential synergistic benefits when MSAs are combined with other therapeutic approaches, such as tau aggregation inhibitors or phosphorylation modulators, offering the possibility of multi-target combination therapies that address different aspects of tau pathologies.

### 3.5. Blocking Tau Propagation Across Neural Circuits

Since cell-to-cell propagation of tau is one of the strongest predictors of clinical progression in individuals with neurodegenerative diseases, researching methods to block tau propagation is necessary [171]. The propagation cascade comprises three interconnected phases: neuronal secretion, extracellular transport, and recipient cell uptake, each presenting distinct therapeutic intervention opportunities. Mechanistically, these interventions target the exosome export and receptor-mediated uptake (HSPG/LRP1) steps that control seed availability and neuronal entry, respectively.

Pathological tau is exported from neurons via unconventional pathways that bypass the classical ER–Golgi route. Activity-dependent release at presynaptic terminals creates focal spots for seed export, and late endosome and lysosome routes regulated by soluble N-ethylmaleimide-sensitive factor attachment protein receptors (SNAREs) (e.g., VAMP8) further expand the extracellular pool. VAMP8 upregulation increases tau secretion while lowering intracellular accumulation. Thus the inhibition of nSMase2 with GW4869 suppresses exosome biogenesis and reduces extracellular seed availability, thereby lowering the input flux of propagating species [172]. Once released, tau spreads as free assemblies and within extracellular vesicles, which stabilise propagation seeds and distribute them along connected circuits. This interstitial reservoir scales with neuronal activity and glial handling, positioning vesicle biogenesis and release as tractable control points. Extracellular seeds engage specific cell-surface receptors that govern internalisation efficiency [173]. Heparan-sulfate proteoglycans (HSPGs) via sulfation-pattern-dependent binding trigger macropinocytosis, while low-density lipoprotein receptor-related protein 1 (LRP1) supports receptor-mediated endocytosis of monomeric and aggregated tau, disrupting HSPG interactions or antagonising LRP1-mediated endocytosis limits neuronal entry and downstream seeding [174]. After endocytic uptake, endolysosomal escape permits access to cytosolic tau and templated misfolding, sustaining the prion-like spread across anatomically connected regions [175,176]. Because extracellular neutralisation is a major choke point, epitope selection in tau-directed immunotherapy remains a key determinant of intercepting efficiency and routing immune complexes to clearance pathways.

### 3.6. Mechanistic Integration and Combination Therapies

Different treatments for tauopathies correspond to different steps of tau pathobiology. From a mechanistic perspective, combination therapy is most appropriate when the pharmacodynamic antagonism is limited and safety interactions are manageable [177]. One of the strategies is the combination of tau-targeting ASOs and tau immunotherapy.

ASOs directly reduce neuronal tau production, reducing the intracellular conformer supply and the subsequent extracellular propagation, while passive or active immunotherapy neutralises residual extracellular assemblies and promotes microglial clearance [146,178]. This combination addresses both the source and the vector of proliferation. In clinical applications, it is important to note that strong upstream inhibition may reduce cerebrospinal fluid antigens, thereby reducing the efficacy of standard immunotherapy. Therefore, sequential therapy administering ASO first, followed by the antibody, may be a better approach. Evaluating of efficacy should also be combined with tau-PET results, rather than relying only on fluid biomarkers [179]. Although there is no ASO with concurrent anti-tau antibody treatment to date, the antibody–oligonucleotide conjugate technology and the cross-pathology combination of lecanemab with E2814 provide a reference for future ASO and anti-tau combinations [180,181]. Modulation of the kinase axis can be mechanistically complementary to immunotherapy by reducing phosphorylation and aggregation propensity [182]. Dosage in this strategy should be carefully selected to avoid overcorrecting physiological PTMs that support MT dynamics. Aggregation modulators can be combined with antibodies to prevent new oligomer formation while simultaneously clearing existed pathogenic proteins [183]. However, rapid disaggregation may transiently elevate soluble oligomer levels and expand the pool of antibody targets, potentially leading to inflammation [184]. This again suggests that staggered dosing and tracking of soluble oligomers and localised tau protein PET should be performed in conjunction with biomarker monitoring. MT stabilisers offer a countermeasure at the distal level of phenotypic activity by restoring cytoskeletal stability and axonal transport. Their mechanistic synergy with kinase inhibitors has the potential to reduce tau dissociation and enhance MT stability, but overstabilization can inhibit dynamic instability and impair transport, particularly when combined with PTM blockers [185]. If combining MT stabilisers with kinase inhibitors is considered, CNS selectivity and safety/efficacy endpoints focused on transport should be prioritized.

### 3.7. Biomarker-Guided Stratification and Therapeutic Monitoring

Reliable biomarkers are crucial for the design and interpretation of tau-targeted trials, whether used for patient stratification or monitoring efficacy. Tau-PET imaging has revolutionized the in vivo assessment of tau pathology by enabling regional quantification and staging paralleling Braak progression. Compared to ^18^Fflortaucipir, ^18^F-MK-6240 offers enhanced cortical contrast and higher specificity [186]. In progressive supranuclear palsy and corticobasal syndrome, ^18^F-PI-2620 demonstrated distinct uptake patterns consistent with quadruple tau pathology, supporting its use in stratifying non-AD tauopathy [187]. These imaging modalities are now routinely included in trial inclusion criteria and serve as quantitative endpoints for longitudinal monitoring.

CSF biomarkers remain the most widely validated fluid markers. Phosphorylated tau species, particularly pTau181, pTau217 and pTau231, provide sensitive measures of abnormal phosphorylation upstream of aggregation and are strongly associated with amyloid status and cognitive status [188]. More recently, the MT-binding region fragment MTBR-tau243 has emerged as a highly specific marker of neurofibrillary tangle pathology, closely correlating with tau-PET signals and disease severity [95]. These advances enable CSF panels to simultaneously capture early phosphorylation changes and downstream aggregate burden, providing complementary readouts for treatment monitoring.

Blood biomarkers offer a more convenient alternative to large-scale screening and repeated sampling. Plasma pTau181 and pTau217 are highly predictive of amyloid PET positivity and cognitive stage transition, with pTau217 being more sensitive for earlier changes [189]. Ultrasensitive assays such as Simoa^®^ Tau are enabling the increasing use of blood-based testing in stratified trial workflows, starting with blood screening followed by CSF and confirmatory tau-PET testing to ensure accurate stratification [190].

While biomarker responses are encouraging, their correspondence to clinical benefit varies across programs. Regarding approaches to reduce tau production, the antisense oligonucleotide BIIB080 (IONIS-MAPTRx) dose-dependently reduced CSF total tau and p-tau levels and demonstrated exploratory reductions in tau-PET signals in a randomised phase Ib study in mild AD [191]. This clearly demonstrates proximal target engagement, but formal clinical endpoints will not be determined until results from an ongoing phase II clinical trial are available, so no claim can be made at this time about a correlation between biomarkers and clinical efficacy. Antibodies such as semorinemab and gosuranemab have demonstrated neither clinical efficacy nor consistent biomarker improvements [157,158]. In contrast, E2814 significantly reduced CSF MTBR-tau243 and pTau217 levels and stabilised tau-PET signals, but cognitive outcomes remain awaited [181]. These experiences highlight that biomarker changes do not necessarily translate into functional improvement, and that trials must match pharmacodynamic markers to the mechanisms they are intended to reflect.

Together, tau-PET, CSF phospho-tau biomarkers, and plasma assays have become indispensable for tau-targeted therapeutic development. They allow selection of patients most likely to benefit, enable monitoring of drug engagement with tau biology, and provide early readouts that can guide adaptive trial design. Incorporation of these multimodal biomarkers has already improved the interpretability of recent ASO and antibody studies, and will be critical in future precision-medicine approaches to Alzheimer’s disease and primary tauopathies.

### 3.8. Emergent Therapies and Direction

While current tau-targeting therapies primarily focus on reducing protein levels, inhibiting aggregation, or enhancing clearance, future treatment strategies may encompass more unconventional approaches. Beyond directly lowering tau levels, modulating non-neuronal contributors has emerged as a complementary strategy. Glial cells, particularly microglia and astrocytes, influence the tau propagation and neuroinflammation, suggesting that indirect interventions on the neuroimmune environment may modify disease progression [192]. For example, mesenchymal stem cells could be transplanted to downregulate proinflammatory mediators and active microglial cells, improving the clearance of extracellular tau aggregates and neuronal dendrite growth [193].

Furthermore, targeting the conformation and subtype-specific characteristics of tau is another promising therapeutic approach. Several studies have demonstrated that tau protein forms distinct, disease-specific pathological conformations in different tauopathies like AD and PSP [68]. Thus, future drug design may shift from broadly targeting total tau proteins to developing drugs that can specifically recognise and bind to a particular pathological conformation. For example, molecules could be designed to bind to specific pathological conformations, thus preventing further aggregation of normal tau protein. The challenge with this approach lies in accurately screening and developing highly specific molecules. While current research suggests that conformation-specific nanobodies can recognise oligomeric tau, further studies are needed to determine whether they can alleviate disease progression [194].

Finally, combining tau-targeting therapies with non-pharmacological interventions to achieve synergistic effects may be a promising future direction. This includes combining conventional medications with non-invasive neurostimulation techniques, such as transcranial magnetic stimulation (TMS) or transcranial direct current stimulation (tDCS). These techniques can modulate neuronal circuit excitability, which theoretically can influence tau protein release [195]. Thus, combining pharmacological treatment with specific lifestyle interventions, such as high-intensity exercise or cognitive training, may enhance therapeutic efficacy by activating endogenous neuroprotective mechanisms [196]. This comprehensive treatment approach acknowledges the complexity of tau pathology and aims to achieve better clinical outcomes through multi-dimensional, multi-target interventions, which may represent a more effective strategy for altering the disease progression.

## 4. Overview of Development Status and Pipeline Statistics

The following table (Table 1) presents tau-targeting candidates currently under clinical trials. Entries are sourced from ClinicalTrials.gov and recent literature, with corroboration from company communications and conference materials where available. Candidates are organised by therapeutic class and stage of development to provide a concise view of ongoing efforts across phase I, phase II, and phase III. The majority of pharmaceuticals targeting tau therapy are focused on small molecules (Figure 1), and they account for the highest proportion of medicines in ongoing phase III clinical trials for tauopathies. An increasing number of clinical trials are underway to evaluate potential disease-modifying therapies for tauopathies, with AD being the most well-characterised. These trials cover a wide variety of therapeutic modalities, including small molecules, mAbs, vaccines, ASOs, small interference RNAs (siRNAs), protein-based agents, and natural extracts [13,191,197]. Here, we summarise the current landscape of clinical-stage drugs based on their modality and development phase.

**Figure 1 cells-14-01506-f001:**
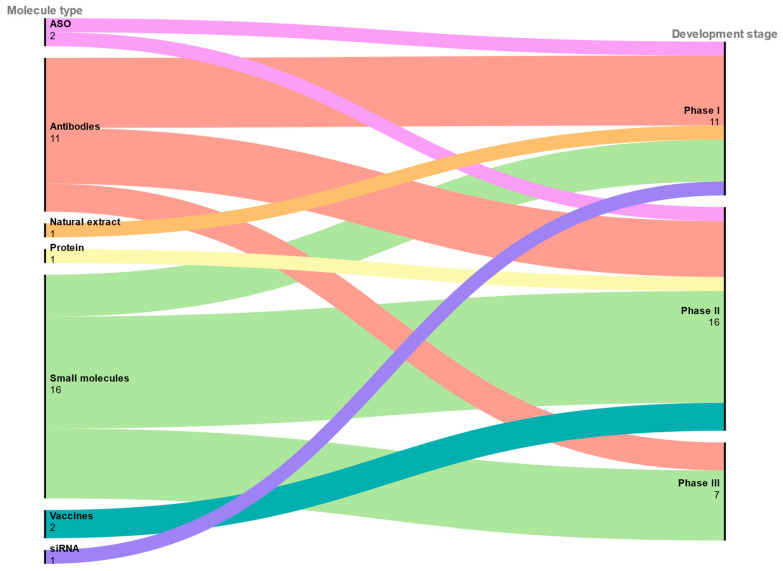
Overview of tau therapeutics in clinical trials: from mechanistic class to clinical outcome (2015–2025). This image was created with RAWGraphs 2.0 (DensityDesign Research Lab, Milan, Italy) [198].

Small molecules comprise the largest class of these pipeline drugs, reflecting their favourable pharmacokinetic properties, including oral availability and efficient BBB penetration. These agents act through diverse mechanisms such as inhibition of tau kinases, prevention of tau aggregation, modulation of neuroinflammation, and enhancement of autophagy [130,131]. Multiple preclinical studies have demonstrated that upregulating autophagy, either pharmacologically or through gene manipulation, can reduce tau burden and rescue phenotypes. Drugs that modulate mTOR-dependent or mTOR-independent pathways (e.g., rapamycin) have repeatedly reduced tau aggregates and improved neurobehavioral outcomes in models [199]. Unlike many tau kinase inhibitors currently in phase II and III clinical trials, small molecule drugs that enhance autophagy have limited clinical validation in tauopathies, and the late autophagosome–lysosome fusion defects observed in human Alzheimer’s disease tissues may complicate a simple autophagy upregulation strategy [200]. Therefore, next-generation approaches (e.g., beclin-1 pathway modulators, nanochaperone systems that target pathogenic tau to autophagosomes) are being explored to restore autophagic flux rather than simply induce autophagosome formation.

mAbs are the second most represented class in the clinical pipeline. These biologics mainly target extracellular tau, which increase the clearance of pathogenic substances through immune-mediated mechanisms. Most mAbs are administered intravenously and are evaluated using fluid and imaging biomarkers, such as CSF phospho-tau and tau-PET ligands. Several antibodies have advanced to Phase II and Phase III studies, reflecting maturing evidence of clinical relevance.

Active immunotherapies targeting tau epitopes are under investigation in early-phase trials. These vaccines are designed to elicit a sustained antibody response, thus achieving long-term therapeutic effects. Although further studies are needed to determine clinical efficacy, early data suggest favourable safety and immunogenicity profiles.

Genetic therapies, including ASOs and siRNAs, offer a mechanism to reduce tau production by silencing the *MAPT* gene. Multiple Phase I clinical trials are investigating ASOs, typically delivered directly into the brain via intrathecal injection. The antisense oligonucleotide BIIB080 (IONIS-MAPTRx) is designed to reduce tau protein by targeting *MAPT* mRNA. In a previous phase I study, intrathecal administration of BIIB080 resulted in dose-dependent reductions in total tau and pTau in CSF with a favourable safety profile. Exploratory imaging studies demonstrated a reduction in tau-PET signal progression, supporting target engagement and biological activity. An ongoing Phase II evaluation (NCT05399888) will determine the durability of tau reduction and whether biomarker changes translate into clinical benefit [191,201]. These therapies are expected to selectively target pathogenic tau without extensively affecting normal function, but long-term effects remain to be clarified [153].

Protein-based approaches, including tau-binding peptides and MT-associated proteins, aim to stabilise cytoskeletal dynamics or prevent tau aggregation [202]. In addition to inhibiting tau production, technologies for catalyzing the removal of intracellular tau through targeted protein degradation have also rapidly advanced. Tau-targeting PROTACs (e.g., C004019) have demonstrated the ability to effectively and selectively reduce pathological tau species in both cell and animal models by recruiting E3 ligases to bind to tau, forming a ternary complex, and promoting ubiquitin–proteasome-mediated clearance [203]. Currently, several medicinal chemistry studies have reported brain-permeable degraders and strategies for optimizing tau selectivity and pharmacokinetics. Molecular glues that stabilise tau-E3 interactions are emerging as a complementary approach [204]. Unlike PROTACs, molecular glues are typically small, monovalent molecules that stabilise the interaction between tau or tau-related proteins and E3 ligases, thereby enhancing ubiquitination and degradation. This design offers a smaller molecular size, potentially improving BBB penetration and oral bioavailability. They can also exploit hidden or shallow binding sites that are not accessible with classic PROTAC linker designs. They can also modulate endogenous tau-ligase contacts without constructing a completely new ternary complex [205]. Application to human studies requires addressing the long-term safety concerns of BBB delivery, the risk of off-target degradation, and chronic regulation of protein homeostasis.

**Table 1 cells-14-01506-t001:** Summary of tau therapeutics in clinical trials (as of July, 2025).

Drug	Molecule Type	Development Stage	Company	Clinical Trial ID	Condition	Indication	Route of Administration	Mechanism of Action	Reference
**ANAVEX2-73**	Small Molecule	Phase III	Anavex Life Sciences Corp. (NY, USA)	NCT04314934	Completed	AD	Oral	A sigma-1 receptor activator that reduces cellular oxidative stress and restores autophagy to enhance cellular homeostasis.	[206]
**Buntanetap (ANVS-401; Posiphen)**	Small molecule		Annovis Bio (PA, USA)	NCT06709014 NCT05686044 NCT05357989	Recruiting Completed Completed	AD AD PD	Oral	Under conditions of iron excess, it enhances the interaction of the iron-responsive element-iron regulatory protein 1 (IRE-IRP1) in the 5′ untranslated region (5′UTR) of target transcripts, thereby selectively reducing the translation of APP, tau, and α-synuclein. This coordinated translational inhibition reduces the burden of misfolded proteins in the amyloid, tau, and synuclein pathways, thereby restoring protein homeostasis.	[207]
**Bezisterim (NE-3107, Triolex, HE3286)**	Small molecule		BioVie Inc. (NV, USA)	NCT04669028	Completed	AD	Oral	NF-κB inhibitor	[208]
**LMTX (hydromethylthionine mesylate, TRx 237, HMTM)**	Small molecule		TauRx Therapeutics (Aber, UK)	NCT03446001	Completed	AD	Oral	Tau aggregation inhibitor disrupts tau aggregates by reversing the proteolytic stability of tau proteins.	[209]
**Nilotinib BE**	Small molecule		KeifeRx, LLC (VA, USA)	NCT05143528	Not yet recruiting	AD	Oral	Administered tyrosine kinase inhibitor.	[210]
**Neflamapimod (VX 745)**	Small molecule	Phase II	Cervomed (formerly, EIP Pharma) (MA, USA)	NCT03435861 NCT05869669	Completed Completed	AD LBD	Oral	Intracellular enzyme p38 mitogen-activated protein kinase alpha inhibitor.	[211]
**FNP-223 (ASN90)**	Small molecule		Ferrer (licensed from Asceneuron) (BCN, ESP)	NCT06355531	Recruiting	PSP	Oral	O-GlcNAcase enzyme inhibitor.	[212]
**T3D-959**	Small molecule		3D Therapeutics (NY, USA)	NCT04251182	Completed	AD	Oral	Dual PPARδ/γ agonist.	[213]
**Censavudine (TPN-101, Festinavir)**	Small molecule		Transposon Therapeutics (CA, USA)	NCT04993768 NCT04993755	Unknown Unknown	PSP ALS/FTD	Oral	It is a potent nucleoside analogue reverse transcriptase inhibitor, repurposed to target LINE-1 retrotransposon activity. By blocking LINE-1 reverse transcriptase, it is proposed to attenuate retrotransposition-driven genomic stress and neuroinflammation. In C9orf72-related FTD/ALS, excessive LINE-1 activity has been implicated in the generation of toxic RNA foci and dipeptide repeat proteins.	[214,215]
**Nicotinamide**	Small molecule		Univ. of California, Irvine (CA, USA)	NCT03061474	Completed	AD	Oral	A coenzyme that inhibits Class III histone deacetylases or sirtuins, which reduces phosphorylation of tau.	[216]
**Varoglutamstat (PQ912)**	Small molecule		Vivoryon Therapeutics (MU, DE)	NCT04498650	Completed	AD	Oral	A glutaminyl cyclase inhibitor, which inhibits post-translational pyroglutamyl modification at the N-terminal of substrate proteins.	[217,218]
**LY-3372689**	Small molecule		Eli Lilly (IN, USA)	NCT05063539	Not recruiting	AD	Oral	Protein O-GlcNAcase inhibitor.	[219]
**BEY2153**	Small molecule		BeyondBio Inc. (Daejeon, KR)	NCT06885567	Not yet recruiting	AD	Oral	Blocks both amyloid and tau.	[220]
**ANAVEX 3–71 (AF710B)**	Small molecule	Phase I	Anavex Life Sciences (NY, USA)	NCT04442945	Completed	AD & FTD	Oral	Targets sigma-1 and muscarinic receptors. It improves cholinergic function and targets the APP metabolism via M1 receptor activation. It provides neuroprotection and anti-amnestic effects via sigma-1 receptor activation.	[221]
**DA-7503**	Small molecule		Dong-A ST Co. (Seoul, KR)	NCT06391827	Not yet recruiting	AD	Oral	Tau aggregation inhibitor, which selectively binds to tau monomers to inhibit tau oligomer formation.	[222]
**OLX-07010**	Small molecule		Oligomerix Inc. (NY, USA)	NCT05696483	Enrolling by invitation	AD & PSP	Oral	Tau self-association inhibitor. It prevents tau oligomers formation and toxicity.	[223]
**BMS-986446 (PRX-005; Anti-MTBR-Tau)**	mAb	Phase III	Bristol-Myers Squibb (NJ, USA)	NCT06268886	Not recruiting	AD	IV	An anti-microtubule-binding region of tau monoclonal antibody,	[224]
**E2814**	mAb		Eisai Inc. (NJ, USA)	NCT05269394 NCT01760005	Not recruiting Recruiting	AD AD	IV	Monoclonal IgG1 antibody, which recognises an HVPGG epitope of tau.	[151]
**AL-101 (GSK-4527226)**	mAb	Phase II	GlaxoSmithKline (partnered with Alector) (LON, UK)	NCT06079190	Not recruiting	AD	IV	A monoclonal antibody to sortilin, which negatively regulates levels of lysosomal protein progranulin leading to neurodegenerative diseases.	[225]
**Posdinemab (JNJ-3657; JNJ-63733657)**	mAb		Johnson & Johnson (NJ, USA)	NCT04619420	Not recruiting	AD	IV	An IgG1/kappa monoclonal anti-phosphorylated tau antibody with high affinity to pT217.	[226]
**Bepranemab (UCB-0107, RG 6416)**	mAb		UCB SA (BXL. BE)	NCT04867616	Not recruiting	AD	IV	IgG4 monoclonal anti-tau antibody binds to amino acids 235-250.	[227]
**ADEL-Y01**	mAb		ADEL Inc. (Seoul, KR)	NCT06247345	Recruiting	AD	IV	Monoclonal antibody targets tau acetylated lysine 280.	[228]
**APNmAb005**	mAb	Phase I	Aprinoia Therapeutics (MA, USA)	NCT05344989	Unknown	AD & tauopathies	IV	Monoclonal antibody that recognises conformational epitope associated with tau oligomers.	[229,230]
**LuAF-87908**	mAb		H. Lundbeck AS (Copenhagen, DK)	NCT04149860	Completed	AD	IV	Monoclonal antibody that targets the C-terminal epitope.	[231]
**MK-2214**	mAb		Merck & Co. (NJ, USA)	NCT05466422	Completed	AD	IV	IgG2a monoclonal antibody targets the phosphorylated tau at Ser 413.	[232]
**VY-7523 (VY-TAU01)**	mAb		Voyager Therapeutics (MA, USA)	NCT06874621	Recruiting	AD	IV	A mAb inhibit tau propagation.	[233]
**Bepranemab (UCB-0107, RG 6416)**	mAb		UCB SA (BXL. BE)	NCT04658199	Not recruiting	PSP	IV	IgG4 monoclonal anti-tau antibody binds to amino acids 235-250.	[234]
**Tertomotide (Riavax, GV1001)**	Vaccine	Phase II	GemVax & KAEL Co Ltd. (Daejeon, KR)	NCT05189210 NCT06235775 NCT05819658	Not recruiting Not recruiting Completed	AD PSP PSP	SC	A 16-amino-acid peptide that inhibits neurotoxicity and apoptosis by mimicking the extra-telomere function of hTERT.	[235,236]
**JNJ-2056 (ACI-35030; JNJ 64042056)**	Vaccine		Janssen Pharmaceutica N.V. (Beerse, BE)	NCT06544616	Recruiting	AD	IM	A liposomal vaccine based on SupraAntigen technology that targets pTau as an anti-tau active immunotherapy.	[237]
**BIIB080 (MAPTRx)**	ASO	Phase II	Biogen (MA, USA)	NCT05399888	Not recruiting	AD	IT	An ASO that targets the mRNA of *MAPT*.	[178]
**NIO-752**	ASO	Phase I	Novartis (Basel, CH)	NCT06372821 NCT05469360 NCT04539041	Not yet recruiting Recruiting Completed	AD AD PSP	IT	An ASO that targets the mRNA of *MAPT*.	[238]
**XPro1595**	Protein	Phase II	INmune Bio (FL, USA)	NCT05522387 NCT05318976	Not recruiting Not recruiting	AD AD	SC	A soluble TNF inhibitor.	[202]
**LY-3954068**	siRNA	Phase I	Eli Lilly (IN, USA)	NCT06297590	Recruiting	AD	IT	A siRNA targets *MAPT*.	[239]
**DDN-A-0101**	Natural extract	Phase I	Pharmacobio (Seongnam, KR)	NCT06367426	Recruiting	AD	Oral	Aggregation inhibitor for tau and APP. And enhance cholinergic function to prevent acetylcholine from being hydrolysed. It was developed based on the gut–brain microbiota axis platform.	[240]

*AD* Alzheimer’s disease; *PD* Parkinson’s disease; *IRE-IRP1* Iron-responsive elements–Iron Regulatory Protein 1; *APP* Amyloid precursor protein; *NF-κB* Nuclear factor kappa-light-chain-enhancer of activated B cells; *LBD* Lewy body dementia; *PSP* Progressive Supranuclear Palsy; *PPAR* Peroxisome proliferator-activated receptor; *ALS* Amyotrophic lateral sclerosis; *FTD* Frontotemporal dementia; *LINE-1* Long interspersed element 1; *M1* Muscarinic acetylcholine receptor 1; *mAb* Monoclonal antibody; *IV* Intravenous; *SC* Subcutaneous; *IM* Intramuscular; *hTERT* Human telomerase reverse transcriptase; *IT* Intrathecal; *ASO* Antisense oligonucleotide; *TNF* Tumor necrosis factor.

Natural compounds with proposed anti-inflammatory or antioxidant properties are being evaluated in formal clinical trials. Although preclinical data suggest potential neuroprotective effects, the low specificity of these drugs is a persistent problem [153].

## 5. Analysis of Discontinued Programs

The subsequent table (Table 2) summarises tau targeting candidates that were discontinued after clinical trials. Information is derived from ClinicalTrials.gov, published reports, and company disclosures. Each entry is organised by modality and by the phase reached at discontinuation, offering context for the challenges encountered in clinical development and for the interpretation of the current pipeline. In recent years, the only monoclonal antibody that has been terminated in clinical trials for tau is Semorinemab (Table 3). Semorinemab is an IgG4 antibody designed to target the N-terminal portion of all six isoforms of phosphorylated and oligomerized tau in treating prodromal-to-mild AD [241]. In preclinical trials, it demonstrated neuroprotective effects in a neuronal and microglial co-culture cell model, as well as tau pathology reduction in IgG2a wild-type and TauP301L-Tg mouse models. The safety of Semorinemab was evaluated in cynomolgus monkeys [242]. Although Semorinemab did not show relevant safety concerns based on the results of the phase I clinical trial, its common primary endpoint did not improve significantly in the subsequent clinical phase II trial, so this project was terminated after the end of the clinical phase II trial [157,241,242]. For prodromal–mild AD individuals, Semorinemab failed to slow clinical progression and did not reduce tau-PET signal over 73 weeks, which suggests that target engagement was insufficient or not disease-modifying at this stage. In mild-to-moderate AD brains, the antibody produced a statistically significant benefit on ADAS-Cog11 but missed the co-primary functional endpoint and demonstrated no effect on secondary measures such as CDR-SB [158]. Though Semorinemab is designed to intercept transsynaptic spread, it has largely failed to address the problem of intracellular pathogenic tau protein species and their downstream aggregation. Mechanistically, the N-terminal epitope may not be sufficiently enriched in pathology, resulting in the problem of intracellular, aggregation-competent tau protein species and their downstream assembly being largely unaddressed, which is a limitation of anti-tau antibodies [243]. Furthermore, the N-terminal domain of tau provides spacing between MTs, and this activity may have an impact on the actual effectiveness of the antibody [244]. As an antibody targeting tau, Semorinemab binds to all tau protein isoforms and cannot bind only to pathological tau proteins, lacking sufficient specificity, which is an improved direction for Posdinemab and other monoclonal antibodies studies currently underway. In addition to the biological properties of the epitope, the choice of intervention timing is also important. Although there is a biomarker interaction, the late symptomatic stage may be beyond the reversible window. In addition, the substantial antibody level and observation window may not be sufficient to translate the pharmacodynamic effect into functional changes. Therefore, future studies should refocus on epitope specificity, recruiting biomarker-confirmed cohorts earlier, optimizing brain exposure, and use of composite, biomarker-anchored endpoints over longer treatment courses.

Beyond biologics such as mAbs, the large number of small-molecule drugs accounts for the largest proportion of clinical drugs discontinued in recent years. ASN-51 and BIIB113 are both protein OGA inhibitors. Although ASN-51 has not reported preclinical efficacy, its first-generation OGA inhibitor has shown effects on enhancing brain glycosylation, preventing NFT formation, and improving exercise behaviour in preclinical animal models [55]. BIIB113 illustrates similar trends to ASN-51 in preclinical experiments presented at the Biogen AD/PD conference. However, animal models have their limitations, especially in terms of the complexity and heterogeneity of simulating AD [251]. In addition, although OGA inhibitors reported high target occupancy, the target contact rate decreased significantly in the synaptic cleft. The drug distribution gradient decreases step by step from CSF to the intercellular space to the protruding space, even if more than 95% of the target contact is achieved in the CSF, the actual contact rate in the protrusion gap may be only 1–3% [252], which may affect the actual efficacy of OGA inhibitors in patients. The disconnection between occupancy and efficacy highlights that CSF pharmacodynamics may overestimate the functional involvement of neuronal interfaces. Moreover, a single-node strategy of OGA inhibition may not be able to effectively address the redundancy and complexity of AD tau pathology. Therefore, perhaps a single target treatment for OGA is insufficient to address the complexity of AD, and a combination strategy targeting both intracellular and extracellular targets, such as pairing OGA inhibitors with extracellular antibodies to simultaneously reduce the formation of new seeds and intercept their propagation, may be a more effective approach [253].

Simufilam (PTI-125) is a small-molecule drug targeting the altered conformation of filamin A, which is an actin-binding scaffolding protein and regulator of the actin cytoskeleton [254]. In AD pathology, Aβ induces a conformational alteration of filamin that stabilises its association with α7 nicotinic acetylcholine receptors (α7nAChRs), thereby enhancing Aβ-driven activation of downstream kinases such as ERKs and JNK1 [255]. This cascade promotes hyperphosphorylation of tau, which in turn impairs its normal MT-stabilising role and drives its redistribution into the somatodendritic compartment, fostering synaptic dysfunction and loss [256]. Furthermore, filamin A overexpression augments tau phosphorylation and cleavage, increasing tau accumulation in neuronal cells [257]. It has been reported that by reducing the levels of FLNA-α7nAChR and TLR4 binding, PTI-125 significantly reduced Aβ42-induced tau phosphorylation and inflammatory cytokine levels. In addition, studies in animal models demonstrated that treatment with Simufilam for two months reduced receptor dysfunction and improved synaptic plasticity in triple transgenic (3xTg) mice (harbouring APP Swedish, *MAPT* P301L, and *PSEN1* M146V mutations) as well as in wild-type mice with mild neuropathology. Behavioural performance in 3xTg mice was significantly improved [258,259]. From 2017 to 2022, both phase I and II clinical trials of Simufilam were completed. According to published data, in clinical phase IIa and IIb, the levels of CSF total tau phosphorylated tau, neurofilament light, neurogranin, and YKL-40 were all reduced after treatment [260]. However, the results of the Phase III clinical data released in 2024 reported that for mild and moderate patients, no significant differences were observed after Simufilam treatment, the program was terminated [245]. Although the regulation of downstream pathways by filamin A has been demonstrated in preclinical studies, it has not been fully validated in clinical studies. The effects of filamin A on tau may be related to brain regions in early disease, which may pose difficulties in the strategy of treatment targeting filamin A [254]. When applied in the late symptomatic phase, pathway recalibration may not be sufficient to reverse established network malfunctions. Second, the effects of FLNA on tau appear region-dependent early in the disease, suggesting that therapeutic efficacy may vary by region [261]. Furthermore, if patients are underrepresented in tau activity, it may be possible to recruit patients with heterogeneous biology in which FLNA anchoring mechanisms are less dominant [257]. Taken together, these findings suggest that future FLNA-targeting strategies should be implemented earlier in the disease phase, combined with agents that more directly reduce tau sources or transmission, and recruit participants with enriched tau pathology.

In addition to O-GlcNAcase inhibitors, some small molecules with other mechanisms have also been developed, such as modulating orai calcium channel activity, inhibiting advanced glycation endproducts, and inhibiting glutaminyl. Although ReS-19T, Azeliragon, and PQ912 have obtained convincing preclinical data, none of them have been able to translate into meaningful clinical benefits. ReS-19T is a small molecule that can restore calcium homeostasis. In the tau pathology model, aberrant tau accumulation leads to the uncontrolled activation of storage and manipulation calcium channels by remodelling the cell cortex through the septin filament. ReS-19T can bind to septin, restore filament assembly in disease states, and inhibit the entry of calcium ions into store-operated calcium channels (SOCCs). In preclinical mouse disease models, ReS19-T restored prominent palpability and mitigated the pathological development of Aβ and tau [249]. Phase I clinical results for ReS19-T demonstrated excellent brain exposure and safety at high doses. However, this dose caused serious adverse effects in phase IIa, with recurrent elevated transaminases in subjects showing hepatotoxicity at high doses [262]. While this does not imply a failure of this therapeutic strategy, the safety window for compounds needs to be further optimised. Based on this, the second-generation drug candidate REM392 has been improved in terms of safety and pharmacokinetics, and is currently undergoing phase I safety experiments [263].

Azeliragon is an antagonist of the receptor for advanced glycation end products (RAGE), which binds ligands such as AGEs and is implicated in inflammatory and vascular responses [264]. In a phase II trial, the low-dose group of Azeliragon demonstrated significant improvement in ADAS-Cog scores among participants with mild AD, which led to initiation of a phase III trial. However, the phase III trial was ultimately terminated because the primary endpoint was not met [247]. Several factors may explain this discontinuation, including Azeliragon’s narrow therapeutic window, limited mechanistic efficacy, heterogeneity in patient populations, and the choice of treatment duration as an endpoint. Moreover, the dose administered may not have been sufficient to achieve a clinically meaningful effect size [265]. Another challenge is that robust cognitive differences in mild AD often require at least 18 months of observation, whereas the 6-month changes in ADAS-Cog are highly variable and frequently fail to reach statistical significance [266].

PQ912 is an oral small molecule glutamine cyclase inhibitor that inhibits glutaminyl cyclase (QC) [267]. By inhibiting the N-terminal pyrrole glutamate upstream of Aβ, the highly aggregated, seed-like pyroglutamate-Aβ (pGlu-Aβ) can be reduced, and the oligomer toxicity and subsequent plaque maturation can be weakened [217]. At the same time, inhibition of glutaminyl-peptide cyclotransferase-like (QPCTL) can reduce the pGlu maturation of chemokines such as CCL2, thereby down-regulating the driver of inflammation and indirectly affecting the tau pathway and network function [268]. Although PQ912 demonstrated a high inhibitory effect on QC activity in clinical IIa studies [77], it failed to meet the primary and secondary clinical endpoints in IIb, resulting in the termination of its development [269]. One possible explanation is that PQ912 suppresses the formation of pGlu-Aβ while leaving pre-existing plaques and higher-order aggregates unaddressed [270]. Accordingly, any clinically meaningful effect size may require a longer exposure before separation on clinical endpoints becomes detectable. In addition, the PQ912’s anti-inflammatory activity indicates that its mechanism is not exclusively centred on central Aβ biology. Although such systemic actions may favour inflammation-related peripheral outcomes, they are unlikely to translate into short-term improvements on core AD cognitive and functional scales. Therefore, extending the observation window or combining PQ912 with a plaque-clearing therapy may be beneficial to convert the mechanistic effects into clinically significant benefits [271]. For Azeliragon (RAGE inhibition) and PQ912 (QC inhibition), the gap reflects a combination of factors: phase mismatch (intervention after widespread network malfunction), patient heterogeneity, and endpoint misalignment (i.e., adequate CSF signaling but lack of synaptic engagement or too short an observation window to capture downstream functional changes) [218,244]. Therefore, future programs may benefit from integrating synaptic exposure metrics with dose selection, extending treatment durations and using biomarker-anchored composite endpoints, or considering pairing intracellular modulators with extracellular seed neutralization approaches to address the multi-node nature of tau pathology.

Based on lessons learned from discontinued programs, the most promising tau-targeting strategies are those that have the ability to address recurring modes of failure, targeting insufficient brain exposure, insufficient selectivity for disease-relevant tau species, and intervention too late in the disease course, while incorporating reliable readouts of target engagement. The most direct approach to reducing tau expression is through intrathecal administration of the ASO BIIB080 (IONIS-MAPTRx), which directly reduces neuronal tau upstream of conformational heterogeneity and intercellular spread [178]. In a randomised Phase 1b trial in mild AD, BIIB080 produced dose-dependent, durable reductions in total and phosphorylated tau in the cerebrospinal fluid with a favourable safety profile, providing pharmacodynamic evidence that early failure is not inevitable through better proximal engagement of the mechanism [191]. Furthermore, next-generation active immunisation seeks to achieve durable, pathology-focused immunity at an earlier stage. The pTau-selective vaccine JNJ-2056 demonstrated robust immunogenicity against phosphorylated tau epitopes in early clinical trials. A post hoc analysis of the related AADvac1 program in Phase 2 showed sustained reductions in neurodegenerative and inflammatory biomarkers in biomarker-confirmed subgroups, supporting the use of enriched, stage-specific trial designs rather than targeting late-stage, clinically heterogeneous populations [161]. Furthermore, to address dosing challenges, platforms that can cross the BBB are crucial. A transferrin receptor “brain-shuttle” construct has been demonstrated to significantly increase CNS exposure and rapidly clear amyloid plaques in humans, validating receptor-mediated transcytosis as a treatment-independent clinical approach, potentially applicable to tau biologics. Magnetic resonance-guided focused ultrasound has been demonstrated to temporarily open the BBB and enhance antibody efficacy in patients, providing a regional and reproducible delivery aid [272]. Because pathogenic tau exerts much of its toxicity intracellularly and propagates through receptor-mediated uptake, intracellular drugs such as AAV-expressed intrabodies or nanobodies have demonstrated the ability to block neuronal internalisation and inhibit seeding of active tau by competing with the LRP1 uptake pathway, thereby addressing the “intracellular blind spot” of early extracellular IgG [273]. These strategies are aligned with the biomarker framework and minimise factors that may affect efficacy, thereby enabling early intervention and exposure-response modelling while reducing translational risk.

## 6. Conclusions

Tau is increasingly recognised as a key driver of neurodegeneration in AD and related tauopathies. Although advances in understanding tau biology have revealed multiple therapeutic opportunities, clinical translation remains challenging. Numerous programs have failed due to intracerebral delivery disorders, safe dose limitations, or suboptimal patient selection and trial endpoints, which reflects the complexity of selective regulation of pathological tau. Despite these difficulties, the therapeutic landscape continues to evolve. Diverse approaches, including small-molecule inhibitors, passive and active immunotherapies, and nucleic acid-based strategies, are progressing through clinical pipelines. Moreover, combination strategies that target complementary pathways or simultaneously address amyloid and tau pathology may overcome the limitations of single-target interventions. Importantly, future progress may depend not only on the development of novel drugs but also on early intervention in biomarker-positive individuals to target upstream biological mechanisms before widespread tau deposition occurs. Increasing reliance on biomarker-guided stratification will help enrich patients most likely to respond, while rational combination therapies may mitigate the limitations of monotherapy. Meanwhile, a precision medicine framework that considers genetic background, tau isoform biology, and disease phenotype, coupled with delivery innovations that enhance brain penetration, provides a path to more personalized and effective interventions. With continued innovation in these frontiers and a deeper understanding of tau pathogenesis, tau-targeted therapies may ultimately lead to meaningful disease improvement in Alzheimer’s and other tauopathies.

## Figures and Tables

**Table 2 cells-14-01506-t002:** Summary of tau therapeutics in discontinued clinical trials (as of July, 2025).

Drug	Molecule Type	Last Known Development Stage	Company	Clinical Trial ID	Indication	Route of Administration	Mechanism of Action	Reference
**Semorinemab**	mAb	Phase II	Genentech (CA, USA)	NCT03828747 NCT03289143	AD	IV	Anti-tau mAb acts by targeting misfolded tau.	[158,241,242]
**Simufilam (PTI-125)**	Small-molecule	Phase III	Cassava Sciences, Inc. (TX, USA)	NCT05575076	AD	Oral	It binds to an altered form of the protein filamin A (FLNA)	[132,245]
**Azeliragon (PF 04494700; RAGE antagonist; TTP 488)**	Small-molecule		vTv Therapeutics (NC, USA)	NCT03980730	AD	Oral	RAGE antagonist	[246,247]
**ASN-51**	Small-molecule	Phase II	Asceneuron (VD, CH)	NCT06677203	AD	Oral	Prevents the buildup of toxic tau aggregates by inhibiting O-GlcNAcase.	[248]
**ReS-19T (REM0046127)**	Small-molecule		reMYND (Leuven. BE)	NCT05478031	AD	Oral	It modulates Orai calcium channel activity to restore disrupted calcium homeostasis in neurons.	[249]
**Varoglutamstat (PQ912)**	Small-molecule		Vivoryon Therapeutics (MU. DE)	NCT03919162	AD	Oral	A glutaminyl cyclase inhibitor, which inhibits post-translational pyroglutamyl modification at the N-terminal of substrate proteins.	[217]
**BIIB113**	Small-molecule	Phase I	Biogen (MA, USA)	NCT05195008	AD	Oral	Protein O-GlcNAcase inhibitor.	[250]

*AD* Alzheimer’s disease; *mAb* Monoclonal antibody; *IV* Intravenous; *RAGE* Receptor for advanced glycation end products.

**Table 3 cells-14-01506-t003:** Taxonomy of causes for failure of tau-targeted clinical trials (as of July, 2025).

Drug	PK Issues	Dosing Issues	Endpoint Limitations	Population Issues	Design Flaws
Semorinemab	Suboptimal brain exposure suspected; extracellular target may miss intracellular pathogenic tau	–	Missed co-primary; no slowing of clinical progression; no tau-PET reduction over 73 weeks	Intervention at late symptomatic stages; potential insufficient target engagement in prodromal–mild AD	Observation window may be inadequate; epitope not pathology-enriched; limited specificity to pathological tau
ASN-51	High CSF occupancy but low synaptic interface engagement (CSF → ISF → synaptic gradient)	–	Occupancy–efficacy disconnect (CSF PD may overestimate neuronal interface engagement)	–	Single-node strategy may be insufficient for redundant tau pathology; consider combinations
BIIB113	Similar to ASN-51: limited synaptic engagement despite high CSF occupancy	–	Occupancy–efficacy disconnect	–	Single-node strategy; translational limits of animal models
Simufilam/PTI-125	–	–	No significant benefit on primary endpoints in Phase III	Applied in late symptomatic phase; potential regional dependence and heterogeneous biology with underrepresented tau activity	Pathway recalibration may be insufficient at late stage; mechanism may be region-dependent early
ReS-19T	Excellent brain exposure in Phase I	Dose-limiting hepatotoxicity at higher exposure.	–	–	–
Azeliragon	–	Narrow therapeutic window	Primary endpoint not met; observation duration likely too short (ADAS-Cog changes over 6 months highly variable)	Population heterogeneity (mild AD)	Treatment duration/endpoint alignment suboptimal; limited mechanistic efficacy
PQ912	–	–	Failed primary and secondary endpoints; benefits may require longer exposure	–	Mechanism leaves existing plaques unaddressed; anti-inflammatory systemic actions dilute central effects; may need combination therapy

*PK* pharmacokinetics; *PET* positron emission tomography; *AD* Alzheimer’s disease; *CSF* cerebrospinal Fluid.

## Data Availability

No new data were created or analyzed in this study.

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
