# Peer review of "Tau-Targeted Therapeutic Strategies: Mechanistic Targets, Clinical Pipelines, and Analysis of Failures"

_cells, 2025, doi:10.3390/cells14191506_

Round 1

Reviewer 1 Report

Comments and Suggestions for Authors

Dear Editors, dear Authors,

this is apparently a comprehensive work summarizing important past and present therapeutic approaches. The introduction is precise and gives sufficient introduction to the main theme, therapies, which is well organized and described. I congratulate the authors for this apparently comprehensive (?) work.

I strongly recommend publication of the article, granting major revision, of the following points: -The tables are very important, but not legible, the layout is insufficiently formatted, I suggest to dramatically decrease font size, use the whole width of the page, and even put the table horizontally, or something like this. 

-The review appears exhaustive for tau-based therapies, if so this must be stated clearly, but in any case the search strategies and the limitations of the search strategies, as well the intention/aims should be clearly stated, in the beginning, but also for each section.

-A section of emergent therapies should be included.

Some minor comments:

-Genes are to be italicized

-line 56ff: "tau loses MT affinity and mislocalizes from axons to somatodendritic compartments, where exposed repeat domain motifs confer seeding competence and initiate elongation of cross-β filaments [21]...": Please revise, it has been extensively shown and discussed that it is rather a failure of axonal targeting of Tau rather than axon-to-soma transit that results in somatic presence of tau

-Tau is also expressed in oligodendrocytes, and in disease conditions also elsewhere, please revise and properly address

-Tau has more than 6 isoforms, in fact the canonical MANE transcript is closest to the socalled Big Tau isoform (completely neglected here), please revise and properly address

-line 81ff “It has been implicated in modulating long‑term memory, habituation, sleep‑wake cycles, and synaptic plasticity via effects on long-term potentiation/depression (LTP/LTD) [37–39].”  Please avoid “it” in scientific texts, in the above sentence authors probably refer to Tau, but this is not clear, even if redundant do not use “It, this, etc” lightly but properly indicate what you are referring to, otherwise the article is too difficult to read and too ambiguous

-line 86ff “Tau regulates motor protein dynamics by forming concentrated microdomains along axonal microtubules, thereby differentially affecting kinesin and dynein mobility, enabling precise spatial control and cargo delivery.”: Reference missing

-line 144ff “AD contains mixed 3R or 4R PHFs and NFTs with early pTau181, pTau217 dynamics, while PSP or CBD are 4R-predominant with distinct phosphorylation signatures”: Reference missing

-Lines 307-309: revise sentence/check grammar, not understandable

Reviewer 2 Report

Comments and Suggestions for Authors

This review provides a solid overview of tau-targeting therapeutic strategies, which is is highly relevant to both academic and clinical audiences. Tau pathology is central to multiple neurodegenerative diseases (AD, FTD, PSP, CBD, etc.), and therapeutic approaches are rapidly evolving. The review is highly relevant to both academic and clinical audiences.

However, there are some major concerns on this review article:

1, While the article is comprehensive, it tends to be descriptive rather than critical. For example, immunotherapy trials are summarized, but limitations (e.g., poor brain penetration, off-target immune responses, trial failures) are not discussed in depth.

2, The review lists strategies (aggregation inhibition, kinase inhibition, etc.) but does not connect them mechanistically. For example, how these approaches might complement or conflict with each other. The authors should add more discussion on it. The authors can discuss how multiple therapeutic strategies might be combined (e.g., tau immunotherapy + ASO; kinase inhibitor + microtubule stabilizer).

3, Several high-profile tau immunotherapy and kinase inhibitor trials have failed or shown modest effects. The review understates the lessons learned from these failures (e.g., timing of intervention, patient selection, disease stage). The authors should provide balanced evaluation of successes and failures, with mechanistic explanations.

4, Emerging areas like PROTACs, autophagy enhancers, and RNA-targeted therapies are briefly mentioned. This article could benefit from more detailed discussion on these aspects. The authors can discuss more and include more detail on ASOs in clinical development (e.g., BIIB080/IONIS-MAPTRx), tau degraders (PROTACs, molecular glues), and autophagy-based strategies. 

5, The review does not sufficiently address how the field should move forward. For examples, the discussion on combination therapies, biomarker-guided stratification, or precision medicine approaches, which can be applied in the future. The authors can end with a roadmap section outlining future priorities: early intervention, patient stratification, combination therapies, personalized medicine.

6, While clinical trials are well-summarized, there is less focus on preclinical insights (e.g., iPSC-derived neurons, organoid models, single-cell omics), which are shaping the future of tau drug discovery. The authors should discuss on it in details.

7, Some sections feel too general (for example, kinase inhibitors, microtubule stabilizers), without a critical comparison of lead compounds. The authors should add extensive discussion in it. The authors should highlight why certain strategies (e.g., kinase inhibitors) have historically failed and how newer approaches are addressing these issues.

8, The authors can expand discussions on tau PET imaging, CSF/phospho-tau biomarkers, and plasma assays as key for trial stratification and therapeutic monitoring.

9. the authors can include summary tables of compounds in clinical trials with outcomes, highlighting those discontinued vs ongoing.

Reviewer 3 Report

Comments and Suggestions for Authors

This is a timely and ambitious synthesis of tau-targeted therapeutic strategies, covering tau biology from post-translational modifications to pathology propagation and therapeutic approaches. The manuscript is well structured, with particularly strong sections on enzymatic modulation, aggregation inhibitors, and immunotherapies. The inclusion of detailed tables summarizing both active and discontinued clinical programs is especially valuable, connecting mechanistic insights with clinical outcomes and offering a panoramic view of the field. Overall, this review represents a constructive contribution for both basic researchers and clinicians working on tauopathies.

However, I would like to notify my concerns to the authors: 

  • Transparency and reproducibility: Narrative review without explicit methods (databases searched, inclusion/exclusion criteria, verification of trial data).

  • Quantitative synthesis limited: Claims such as “largest class” or “majority of drugs” lack aggregated numbers or visual support.

  • Trial data inconsistencies: Some discrepancies between text and tables regarding program status (e.g., BIIB113).

  • Failure analysis selective: Focuses on a subset of discontinued trials without clear criteria for inclusion.

  • Weak biomarker–clinical linkage: Biomarker changes (e.g., CSF p-tau reductions) are reported but not consistently tied to clinical endpoints or effect sizes.

  • Mechanistic detail uneven: Certain agents (e.g., ANVS-401, Censavudine) described only broadly, with limited molecular depth.

  • Tau heterogeneity underexplored: Limited discussion on isoform diversity (3R/4R), strain variation, and compartmentalization in relation to epitope targeting.

  • Combination strategies underdeveloped: Mentioned briefly in the conclusion without concrete, stage-specific proposals.

  • Editorial and usability issues: Minor errors (e.g., “MBTR” instead of MTBR), references of questionable relevance, and dense tables without visual aids for quick pattern recognition

And here are my suggestions to try to improve the quality of the manuscript: 

  • Add a methods section describing search strategy, databases, time windows, and inclusion/exclusion criteria to improve transparency.

  • Provide quantitative synthesis and visuals (e.g., bar charts of programs by modality, Sankey diagrams, timelines) to support narrative claims.

  • Clarify trial data by cross-checking tables and text, and add footnotes explaining program status (e.g., completed vs. discontinued).

  • Systematize failure analysis with a taxonomy of causes (PK, dosing, endpoints, population, design) and present in a matrix or table for clarity.

  • Strengthen biomarker–clinical links by including effect sizes, confidence intervals, and whether biomarker shifts correlated with cognitive or functional outcomes.

  • Expand mechanistic detail for underdeveloped strategies, linking molecular pathways more explicitly to tau pathobiology.

  • Deepen discussion of epitope/isoform heterogeneity and its impact on antibody, ASO, and vaccine design.

  • Introduce a section on combination strategies, outlining rational, stage-specific schemas (e.g., early OGA inhibition, mid-stage anti-tau + anti-Aβ).

  • Correct editorial issues (e.g., MTBR typo, reference checks) and improve table usability by adding abbreviations, legends, and possibly a “Key Messages” summary figure.

Round 2

Reviewer 1 Report

Comments and Suggestions for Authors

Dear Authors, thank you for revising the work. I would like to request add to carefully go through the manuscript and pay particular attention to the references, often the references cited do not fully support the meaning of the (whole) sentence, if there are multiple references cited and only some part of the sentences are supported by the sentence just put the reference to the meaning it supports. 

An example is line 109ff, where there are several points that are simply not in the references, so please separate the content of the reference from your implications:  "This physiological baseline becomes clinically actionable once isoform ratios shift in disease, because epitope choice for antibodies and vaccines and splice-modulating strategies for ASOs can be aligned to the predominant isoform pool [35,39]" In the references there is no work done on Asos, vaccines etc, please clearly point this out. 

I obviously cannot do this for all references, but it is clear that this must be checked properly. It is extremely cumbersome if you read an interesting statement in a review, and try to look it up in the specified reference, and the content is simply not there - please critically reflect.

The table is still difficult to read, redo. E.g. in the column "molecule type" there is nothing specified for the Buntanetap, LMTX etc., one must guess that the "small molecule" listed next to anavex2-73 also applies to them. But hey, help the readers out, just specify or reorganize your table. Same for dev. stage, one has to guess that simply from that point onward it is all the same, I guess the author used a column with separating lines (which are omitted in the version I have). Please find a solution here. Restating seems like a good idea if you have to adhere to the cell/MDPI format.

Some minor mistakes like "...the principal tau phosphorylation PP2A is inhibited by..." (line 232) should probably be "phosphatase".

Section on emerging therapies still missing, what would really change the landscape? A paragraph on out of the box would be nice.

Again congratulation, it is a nice piece of work.

Reviewer 2 Report

Comments and Suggestions for Authors

The authors have carefully addressed all reviewers comments and improved the quality of their review. No further comments. Suggest accepting it in the current form. 

Author Response

Dear editor and reviewers:

Thank you for your thoughtful recommendation for acceptance on our manuscript, “Tau-targeted therapeutic strategies: mechanistic targets, clinical pipelines, and analysis of failures” (Manuscript ID: cells-3857842). We appreciate your time and the feedback that helped us improve it.